# Adalimumab Originator vs. Biosimilar in Hidradenitis Suppurativa: A Multicentric Retrospective Study

**DOI:** 10.3390/biomedicines10102522

**Published:** 2022-10-09

**Authors:** Martina Burlando, Gabriella Fabbrocini, Claudio Marasca, Paolo Dapavo, Andrea Chiricozzi, Dalma Malvaso, Valentina Dini, Anna Campanati, Annamaria Offidani, Annunziata Dattola, Raffaele Dante Caposiena Caro, Luca Bianchi, Marina Venturini, Paolo Gisondi, Claudio Guarneri, Giovanna Malara, Caterina Trifirò, Piergiorigio Malagoli, Maria Concetta Fargnoli, Stefano Piaserico, Luca Carmisciano, Riccardo Castelli, Aurora Parodi

**Affiliations:** 1Division of Dermatology, Department of Health and Science (DissaL), Policlinico San Martino Hospital, IRCCS, 16132 Genova, Italy; 2Department of Clinical Medicine and Surgery, Section of Dermatology, University Hospital Federico II, 80131 Naples, Italy; 3Department of Biomedical Science and Human Oncology, Second Dermatologic Clinic, University of Turin, 10124 Turin, Italy; 4Institute of Dermatology, A. Gemelli University Polyclinic, IRCCS and Foundation, Sacred Heart Catholic University, 00168 Rome, Italy; 5Department of Dermatology, University of Pisa, 56126 Pisa, Italy; 6Dermatology Unit, Department of Clinical and Molecular Sciences, Polytechnic Marche University, 60126 Ancona, Italy; 7Department of Dermatology, University of Rome “Tor Vergata”, 00133 Rome, Italy; 8Department of Dermatology, ASST Spedali Civili di Brescia, 25126 Brescia, Italy; 9Section of Dermatology and Venereology, Department of Medicine, University of Verona, 37129 Verona, Italy; 10Department of Biomedical and Dental Sciences and Morphofunctional Imaging, University of Messina, 98122 Messina, Italy; 11Dermatology Unit, Grande Ospedale Metropolitano “Bianchi Melacrino Morelli”, 89124 Reggio Calabria, Italy; 12Dermatology Unit, Azienda Ospedaliera San Donato Milanese, 20097 Milan, Italy; 13Dermatology, Department of Biotechnological and Applied Clinical Sciences, University of L’Aquila, 67100 L’Aquila, Italy; 14Dermatology Unit, Department of Medicine, University of Padova, 35122 Padua, Italy; 15Division of Biostatistics, Department of Health and Science (DissaL), Policlinico San Martino Hospital, IRCCS, 16132 Genova, Italy

**Keywords:** adalimumab originator, adalimumab biosimilar, Hidradenitis Suppurativa

## Abstract

This study aimed to compare adalimumab originator vs. biosimilar in HS patients, and to evaluate the effect of a switch to a biosimilar, or a switch back to the originator, in terms of treatment ineffectiveness. Patients with a diagnosis of HS were enrolled from 14 Italian sites. Treatment ineffectiveness was measured using Hurley score. The major analyses were 1) comparison between the two treatment groups (non-switcher analysis), and 2) the cross-over trend of Hurley score between treatment switchers (switcher analysis). Cox and Poisson regression models were used to compare the treatment ineffectiveness between groups. A total of 326 patients were divided into four groups: 171 (52.5%) taking originator; 61 (18.7%) patients taking biosimilar; 66 (20.2%) switchers; 28 (8.6%) switchers from originator to biosimilar and switched. A greater loss of efficacy was observed in the group allocated to the biosimilar than the originator group. The switcher analysis showed an effectiveness loss in the biosimilar compared to the originator. These results seem to indicate that a switch from one drug to the other may lead to a greater risk of inefficacy. A return to the previous treatment also does not ensure efficaciousness.

## 1. Introduction

Hidradenitis suppurativa (HS) is an inflammatory skin disease with a characteristic clinical presentation of recurrent or chronic painful or suppurating lesions, especially in the apocrine gland-bearing regions [1,2]. The estimated prevalence of the disease varies widely, between 0.05% and 4.1%, with a female-to-male ratio of 3:1. HS typically occurs in young adults and is associated with lower socioeconomic status [3]. Treatment of HS remains challenging. Being a pleomorphic disease, a single therapy is effective, and a holistic approach is mandatory [4,5]. First line treatment options include clindamycin 1% lotion, recommended in mild disease, while tetracycline and a combination of clindamycin and rifampicin are the drugs of choice in moderate and moderate-to-severe disease. Besides antibiotic therapy, adjuvant therapies such as pain management, weight loss, tobacco cessation, and application of appropriate dressings are needed [6,7]. Among biologic therapies, only adalimumab, an anti-tumor necrosis factor (anti-TNF), is approved in the United States and in the European Union for patients affected by moderate to severe HS [8,9].

Besides adalimumab originator, new biosimilar drugs are currently available, and less expensive than the branded version [10,11]. Rigorous studies are needed to demonstrate the bioequivalence (pharmacokinetic similarity), safety, and immunogenicity of candidate adalimumab biosimilars [12,13], and many authors have agreed that switching from originator to biosimilars should generally be considered safe and effective [14,15]. However, multiple switching between different biosimilars and originator is not recommended, and some authors have raised concerns about the efficacy of biosimilars in patients previously treated with adalimumab originator [16]. Further studies are needed, in order to ensure that biosimilars are as effective as the originator in HS.

For this reason, we conducted a study to compare the effects of adalimumab originator and adalimumab biosimilar for the treatment of HS in a real-world setting. Furthermore, the effects of treatment switching, from the originator to biosimilar, were investigated.

## 2. Materials and Methods

### 2.1. Study Design and Population

This was a sponsor-free retrospective multicenter study, in which 14 Italian sites were involved. Patients were selected from March 2020 to June 2020.

Inclusion criteria were as follows: age ≥18 years; patients diagnosed with HS, under treatment with adalimumab for six months. Exclusion criteria were as follows: Age <18 years; inability or unwillingness to provide informed consent.

The present study was conducted in accordance with the Helsinki Declaration of 1975, as revised in 1983, on Ethical Principles for Medical Research Involving Human Subjects, and approved by the San Martino Ethical Committee, with the number 181. All eligible patients provided written informed consent. As the outcome measure of HS severity we used the Hurley score [17].

### 2.2. Timeline

For each patient, the visit at which adalimumab (originator or biosimilar) was prescribed was considered the baseline (T0). The first time point (T1) was the last available follow-up for patients who had not changed prescription or the date of the first treatment switch; the second time point (T2) was the last available follow-up for patients who switched treatment once or the date of the second treatment switch; the third and last time point (T3) was the last available follow-up for those who switched twice. At all time-points, Hurley scores were recorded. The Hurley staging system was created by Hurley to describe HS severity in one skin region prior to surgery, it is widely used as an outcome measure in trials of medical therapies for HS [18].

### 2.3. Statistical Analyses

Two major statistical analyses were performed: (1) a comparison of patients’ Hurley scores, to estimate and compare the length of the treatment effect between the two treatment groups (namely, the non-switcher analysis), and (2) the cross-over trend of Hurley score between treatment switchers (namely, the switcher analysis).

According to our sample size calculation, 172 subjects were required, to have a 90% power of detecting a doubling of risk with an alpha level of 0.05, assuming a ratio of about 1:2 between the two treatments and 0.5 as the square of the correlation between treatments and the other covariates.

Since the choice to prescribe the biosimilar is regulated by non-clinical factors, such as regional or local hospital internal policies, we expected to find small differences between the two treatment groups. A matching algorithm was created, in order to exclude the most different (i.e., incomparable) patients.

Categorical variables were reported with the count and percentages; continuous variables were expressed as mean and standard deviation (SD) and/or median and interquartile range (IQR), according to their distribution. A propensity score method was used to match patients treated with the originator and with the biosimilar in a matching ratio of 2 to 1; to identify the propensity of treatment, the age, sex, BMI, smoking, disease duration, disease location, and Hurley score at baseline were considered. A nearest neighbor matching method was used as the matching algorithm. A Chi squared test was used to test the association between categorical variables, and T-test or Mann-Whitney test were used to compare continuous variables between groups.

The primary endpoint of this study was treatment ineffectiveness, measured as a Hurley score difference of 0 (stable) or an increase of Hurley score (worsening) between the start of treatment and the end or the last available follow-up. In order to account for the different length of observations, the Kaplan Meier method and the Cox proportional hazard model were used for estimations. Results were reported as the hazard ratio (HR) and 95% confidence interval (95% CI). The incidence risk of treatment ineffectiveness and the incidence risk ratio (IRR) between two groups were used to compare subsequent treatments in the same patient using a person–month unit of observation to account for the different follow-up lengths.

*p*-values below 0.05 were considered statistically significant.

R-software version 3.6.3 was used for all statistical analyses (STAT1) [19], using the MatchIt package (STAT2) [20] for matching algorithm implementation.

## 3. Results

A total of 326 patients met the inclusion criteria and were enrolled. Four groups of patients were identified within the cohort. In the first group, 171 (52.5%) patients started and continued with the originator, from baseline to the end of follow-up; in the second group, 61 (18.7%) started and continued with the biosimilar, from baseline to the end of follow-up; in the third group, 66 (20.2%) switched from originator to biosimilar; and, in the last group, 28 (8.6%) switched from the originator to biosimilar and then switched back to the originator, as shown in Figure 1.

### 3.1. Non-Switcher Analysis

After a 2 to 1 match, 174 comparable patients with HS, who started and continued treatment with adalimumab (n = 116 originator; n = 58 biosimilar), were included in the non-switcher analysis, and their baseline characteristics are summarized in Table 1.

Confirming the matching algorithm, no significant differences were detected between patients treated with the originator compared to patients treated with the biosimilar, in terms of age, sex, BMI, smoking, disease duration, HS localization, or baseline Hurley score (Table 1).

The median follow-up was longer for the originator group compared to the biosimilar (13 months (IQR = 12, 23) vs. 10 months (IQR = 5, 12)). The percentage of patients with effective treatment within the originator group was 87.7% (95% CI 81.9–94.0), compared to 77.1% (95% CI 67.0–88.9) within the biosimilar group after 6 months from the treatment start and 82.2% (95% CI 75.4–89.6) compared to 60.5 (95% CI 48.6–75.5) after 10 months, respectively.

Thus, among patients who started a new treatment, about a four-times faster treatment ineffectiveness was detected in the biosimilar group compared to the originator group (HR = 3.8, 95% CI 2.3–6.2; *p* < 0.001). Despite the matching, we also adjusted all baseline clinical variables (age, sex, BMI, location, smoking, and baseline Hurley score) for residual confounding, and the crude unadjusted HR (3.5, 95% CI 2.2–5.6) was minimally changed. Kaplan–Meier curves, together with the number of patients at risk, the number of censors, and the number of events, are reported in Figure 2.

### 3.2. Switcher Analysis

Since the switches were not planned study interventions, no patients assigned to biosimilar were switched to the originator group without a clinical reason, and all considered patients switched from the originator to the biosimilar group. Therefore, all Hurley scores of biosimilar from switchers were collected after the treatment with originator.

We compared the before-treatment effect and after-treatment effect among the 94 patients who switched treatment. Each patient served as control for himself/herself, with a total of 1670 months of observation. The incidence of ineffectiveness was 4.9 per 100 person-months (95% CI 3.6–6.3) for originator compared to 10.7 per 100 person-months (95% CI 8.3–13.5) for biosimilar. Thus, in the biosimilar group a significantly greater loss of effectiveness was observed, compared to the originator group (IRR = 2.2; 95% CI 1.5–3.2, *p* < 0.001).

To mitigate the bias in favor of the originator, which was given as first treatment in the previous comparison, we also observed the Hurley score trend in the 28 patients (29.8% of the switchers) that switched back to the originator because of effectiveness loss after the biosimilar switch.

As shown in Table 2, the Hurley score variation was associated with the treatment (Chi-squared (4) = 18.2, *p* = 0.001). The variation before and after the biosimilar treatment suggests the possibility that, at least some patients who were originator responders could not respond to the biosimilar. At the re-challenge with originator after switching to biosimilar, five (50%) of the nine originator responders at T1 did not respond as well.

Similar results were observed comparing the Hurley scores of the 61 patients taking the biosimilar as a first adalimumab treatment with the 94 patients taking the biosimilar after switching from originator (HR = 1.16; 95% CI 0.73–1.87; *p* = 0.538), regardless of age, sex, BMI, smoking, disease duration, or last Hurley score before the treatment.

Of the 265 patients starting with originator, 171 (64.5%) continued the treatment to the last follow-up, and the remaining 94 (35.6%) switched to the biosimilar. Among switchers, four patients (4.3%) worsened and 31 (33.7%) improved, among non-switchers no patients worsened and 68 patients (39.8) improved (Figure 3).

### 3.3. Final Model

Finally, with the longitudinal data of the matched non-switcher cohort (N = 174) we estimated the overall treatment effect (regardless of the treatment order), considering treatment group as a time-dependent characteristic.

The percentage of patients with effective treatment in the biosimilar group was 77.9% (95% CI 68.0–89.2) after 6 months from the treatment start, and 60.8 (95% CI 49.5–74.6) after 10 months.

Including the switchers in the analysis, the biosimilar treatment effect estimate precision improved and decreased the follow-up length difference between treatment groups (12 months (IQR 7, 16) for biosimilar) compared to the non-switcher only analysis. Patients in the biosimilar group showed about a 2.8-fold increased risk of having ineffectiveness (HR = 2.8; 95% CI 2.0–4.1; *p* < 0.001), regardless of age, sex, BMI, smoking, disease location, duration, or baseline Hurley score.

## 4. Discussion

This study investigated the effect of adalimumab originator and adalimumab biosimilar, and the effect that a switch to a biosimilar or a switch back to the originator, can have on Hurley score in HS patients. To our knowledge, this is the first study comparing adalimumab originator vs. biosimilar efficacy in these patients.

The main result of the study is that a fourfold more rapid loss of efficacy was observed in the group allocated to biosimilar compared to the originator group. The group of patients who switched from originator to biosimilar showed a loss of efficacy that was significantly higher than non-switchers (both originator and biosimilar). This result seems to suggest that, if a patient starts adalimumab, whether originator or biosimilar, it should not be switched. Unfortunately, this commonly happens in clinical practice, given that biosimilars are less expensive than originators.

Comparing a pooled group of patients taking the originator and those who switched to biosimilar, the biosimilar group showed a more than twofold significantly higher loss of efficacy than the originator group. Although it cannot be ascertained that the originator is more efficacious than the biosimilar, these data indicate that a switch from one drug to another leads to a greater risk of inefficacy.

In addition, a return to the previous treatment with originator also does not ensure efficaciousness. As such, in the management of HS, treatment should begin and continue with the same drug.

From these data, the originator seems to have a greater durability than the biosimilar. This could be explained by considering the immunogenicity of the anti TNFα [21,22]. In fact, it is known that neutralizing antibodies can be created during treatment with this class of biological drugs [23,24]. Thus, antibodies could be potentiated in those patients who have undergone the switch, following antigenic stimulation caused by a similar but not homologous molecule [25].

This study presents some limitations. First, as the outcome measure, the Hurley score was used. This indicator is not sensitive to changes and cannot provide a global severity measure. However, it is relatively quick to perform and widely used in HS research and clinical practice [26].

Second, it should be noted that the Hurley score assessed at the time of the switch from originator to biosimilar could be biased by the former treatment. Indeed, the switch was not due to a loss of efficacy, but to economic reasons. Once the patients switched to biosimilar, starting from a lower Hurley score, an improvement was observable.

Third, four groups were observed: the non-switcher originator, the non-switcher biosimilar, those who switched from originator to biosimilar, and those who switched back to the originator. No patient who started with the biosimilar switched to the originator. This could be considered as a limitation; in fact this final option has not been studied.

The strength of the study is its real-world design, therefore reflecting the current Italian situation regarding the choice of HS treatment. Since the choice to prescribe the biosimilar is regulated by non-clinical factors such as regional or local hospital internal policies, these results may have clinical implications. Before starting a biologic therapy for HS, physicians should consider the possibility of prescribing the originator, rather than the less expensive biosimilar, to have a better chance of effectiveness in the long term.

Further studies on this topic are needed, in order to deepen our understanding of these issues.

## Figures and Tables

**Figure 1 biomedicines-10-02522-f001:**
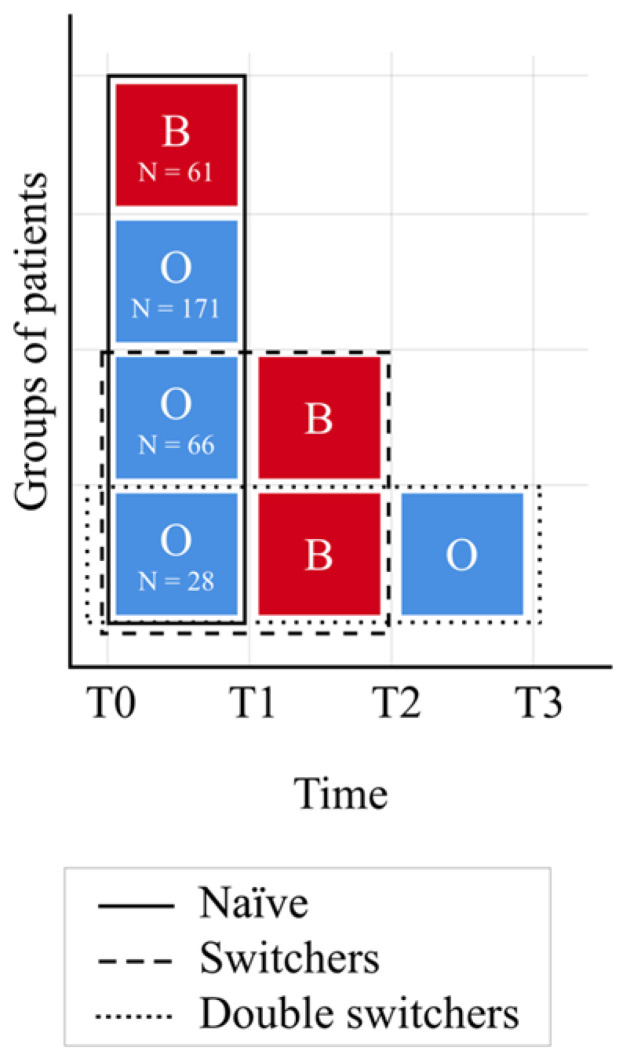
Study groups within the observed cohort of patients. Red B boxes indicate patients taking biosimilar, while blue O boxes indicate patients taking originator. Switch events are presented from left to right.

**Figure 2 biomedicines-10-02522-f002:**
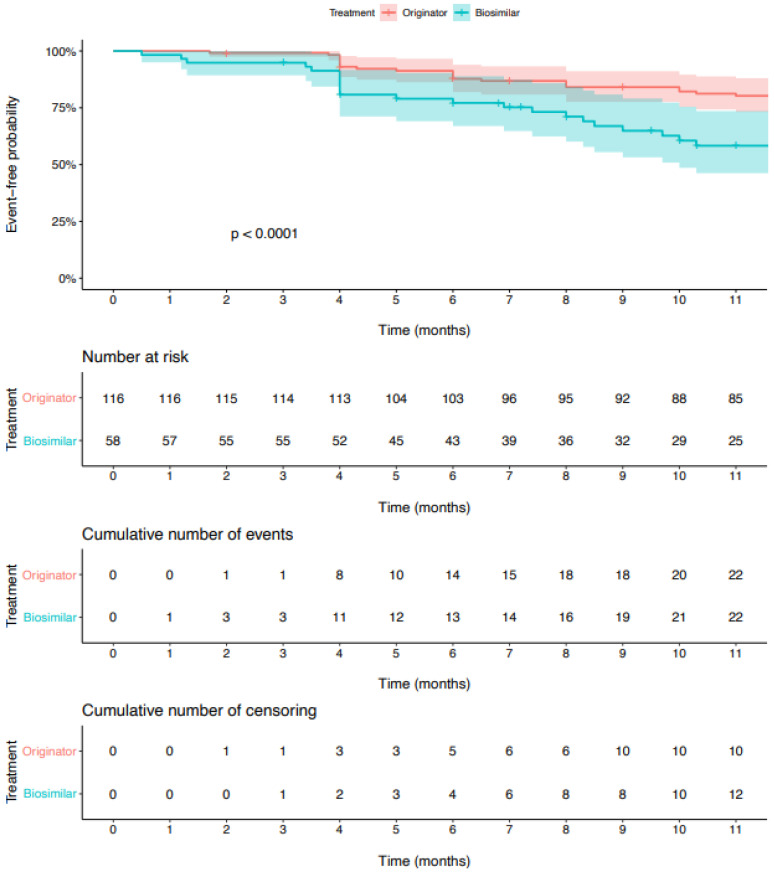
Time to ineffective treatment by treatment group.

**Figure 3 biomedicines-10-02522-f003:**
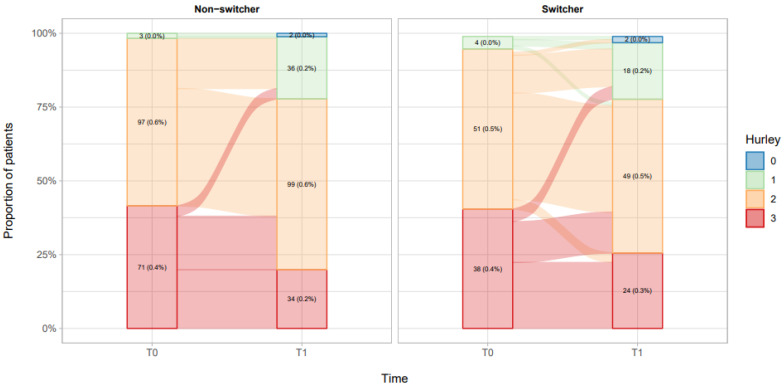
Visual description of Hurley scores among switchers and non-switchers.

**Table 1 biomedicines-10-02522-t001:** Baseline demographic and clinical characteristics of patients included in the non-switcher analysis.

	Overall N = 174	Originator N = 116	Biosimilar N = 58	*p*
**Age (years),** **Mean (SD)**	33.26 (12.13)	33.16 (12.59)	33.45 (11.25)	0.885
**Female sex,** **N (%)**	84 (48.3)	56 (48.3)	28 (48.3)	0.999
**BMI,** **Mean (SD)**	27.22 (4.66)	27.25 (4.68)	27.16 (4.66)	0.910
**Current smoker, N (%)**	129 (74.1)	87 (75.0)	42 (72.4)	0.854
**Disease duration (years),** **Median [IQR]**	10 [5, 19]	10 [5, 19]	10 [5, 19]	0.634
**HS localization, N (%)**				
**Armpits**	78 (44.8)	53 (45.7)	25 (43.1)	0.872
**Groin**	82 (47.1)	53 (45.7)	29 (50.0)	0.707
**Perineum**	21 (12.1)	13 (11.2)	8 (13.8)	0.805
**Others/Unavailable**	74 (42.5)	50 (43.1)	24 (41.4)	0.957
**Hurley score,** **Mean (SD)** **Median [IQR]**	2.37 (0.52) 2 [2, 3]	2.37 (0.52) 2 [2, 3]	2.36 (0.52) 2 [2, 3]	0.915

**Table 2 biomedicines-10-02522-t002:** Hurley score trend of 28 patients with a lack of improvement after switching from originator to biosimilar.

	Hurley Score Worsened, N (%)	Stable Hurley Score, N (%)	Hurley Score Improved, N (%)	*p*
**T1—Challenge (originator)**	1 (3.6)	18 (64.3)	9 (32.1)	0.001
**T2—De-challenge (biosimilar) ***	6 (21.4)	22 (78.6)	0 (0.0)	
**T3—Re-challenge (originator)**	0 (0.0)	23 (82.1)	5 (17.9)	

* 28 patients were selected due to a lack of improvement with the biosimilar.

## Data Availability

The data presented in this study are available within the article. The data that support the findings of this study are available from the corresponding author, upon request.

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
