# Peer review of "Adalimumab Originator vs. Biosimilar in Hidradenitis Suppurativa: A Multicentric Retrospective Study"

_biomedicines, 2022, doi:10.3390/biomedicines10102522_

Round 1

Reviewer 1 Report

1. Introduction needs to be more detailed. 

2. What is the epidemiological data of HS? Refer to the below article.

a. Ingram JR. The epidemiology of hidradenitis suppurativa. Br J Dermatol. 2020 Dec;183(6):990-998. doi: 10.1111/bjd.19435. Epub 2020 Sep 3. PMID: 32880911.

3. What are the current treatments/management for HS?

4. Add a few sentences on biosimilars. Are there any previous reports suggesting that biosimilars are less effective? 

5. There are studies that have reported the bioequivalence, safety and immunogenicity of adalimumab biosimilars. Add a few sentences using some of the published literature.

a. Wynne C, Altendorfer M, Sonderegger I, Gheyle L, Ellis-Pegler R, Buschke S, Lang B, Assudani D, Athalye S, Czeloth N. Bioequivalence, safety and immunogenicity of BI 695501, an adalimumab biosimilar candidate, compared with the reference biologic in a randomized, double-blind, active comparator phase I clinical study (VOLTAIRE®-PK) in healthy subjects. Expert Opin Investig Drugs. 2016 Dec;25(12):1361-1370. doi: 10.1080/13543784.2016.1255724. PMID: 27813422.

b. Gimeno-Gracia M, Gargallo-Puyuelo CJ, Gomollón F. Bioequivalence studies with anti-TNF biosimilars. Expert Opin Biol Ther. 2019 Oct;19(10):1031-1043. doi: 10.1080/14712598.2019.1561851. Epub 2018 Dec 31. PMID: 30574813.

6. In the study design, exclusion criteria should be indicated.

7. Figure-2 and 3 are not clear to read.

8. In the discussion section, the explanation for loss of efficacy due to switching the therapy to a biosimilar is unclear. Can you please write in detail?

Author Response

-1;2;3;4;5 we changed the introduction in this way:

OLD INTRODUCTION

Hidradenitis suppurativa (HS) is an inflammatory skin disease with a characteristic clinical presentation of recurrent or chronic painful or suppurating lesions, especially in the apocrine gland–bearing regions.1,2 Treatment of HS is still challenging. Being a pleomorphic disease, a single therapy cannot be effective, and a holistic approach is mandatory.3,4 In fact, beside antibiotic or biologic therapy, considered as first line therapies, also adjuvant therapies like pain management, weight loss, tobacco cessation, and application of appropriate dressings are needed.5,6  Among biologic therapies, only adalimumab, an anti-tumor necrosis factor (anti-TNF), is approved in the United States and in the European Union for patients affected by moderate to severe HS.7,8

Beside adalimumab originator, new biosimilar drugs are currently available, less expensive than the branded one.9,10 However, it is still unclear whether these biosimilars are as effective as the originator to treat HS.11,12

For this reason, we conducted a study to compare the effects of adalimumab originator and adalimumab biosimilar for the treatment of HS in a real world setting. Further, the effects of treatment switch from originator to biosimilar were investigated.

NEW INTRODUCTION

Hidradenitis suppurativa (HS) is an inflammatory skin disease with a characteristic clinical presentation of recurrent or chronic painful or suppurating lesions, especially in the apocrine gland–bearing regions.1,2 The estimate prevalence of the disease widely varies between 0.05% and 4.1%, with female-to-male ratio of 3:1. HS typically occurs in young adults, and is associated to lower socioeconomic status3. Treatment of HS is still challenging. Being a pleomorphic disease, a single therapy cannot be effective, and a holistic approach is mandatory.4,5 First line treatment options include clindamycin 1% lotion, recommended in mild disease, while tetracycline and a combination of clindamycin and rifampicin are the drug of choice in moderate and moderate-to-severe disease. Besides antibiotic therapy, adjuvant therapies like pain management, weight loss, tobacco cessation, and application of appropriate dressings are needed.6,7 Among biologic therapies, only adalimumab, an anti-tumor necrosis factor (anti-TNF), is approved in the United States and in the European Union for patients affected by moderate to severe HS.8,9

Beside adalimumab originator, new biosimilar drugs are currently available, less expensive than the branded one.10,11 Rigorous studies are needed to demonstrate bioequivalence (pharmacokinetic similarity), safety and immunogenicity of candidate adalimumab biosimilars12,13, and many authors agree that switching from originator to biosimilars should generally be considered safe and effective14,15. However, multiple switching between different biosimilars and originator is not recommended, and some authors have raised concerns about the efficacy of biosimilars in patients previously treated with adalimumab originator16. Further studies are needed in order to assure that biosimilars are as effective as the originator in HS.

For this reason, we conducted a study to compare the effects of adalimumab originator and adalimumab biosimilar for the treatment of HS in a real-world setting. Furthermore, the effects of treatment switch from originator to biosimilar were investigated.

3 Ingram JR. The epidemiology of hidradenitis suppurativa. Br J Dermatol. 2020 Dec;183(6):990-998. doi: 10.1111/bjd.19435. Epub 2020 Sep 3. PMID: 32880911.

14 Wynne C, Altendorfer M, Sonderegger I, Gheyle L, Ellis-Pegler R, Buschke S, Lang B, Assudani D, Athalye S, Czeloth N. Bioequivalence, safety and immunogenicity of BI 695501, an adalimumab biosimilar candidate, compared with the reference biologic in a randomized, double-blind, active comparator phase I clinical study (VOLTAIRE®-PK) in healthy subjects. Expert Opin Investig Drugs. 2016 Dec;25(12):1361-1370. doi: 10.1080/13543784.2016.1255724. PMID: 27813422.

15 Gimeno-Gracia M, Gargallo-Puyuelo CJ, Gomollón F. Bioequivalence studies with anti-TNF biosimilars. Expert Opin Biol Ther. 2019 Oct;19(10):1031-1043. doi: 10.1080/14712598.2019.1561851. Epub 2018 Dec 31. PMID: 30574813.

16 Montero-Vilchez T, Cuenca-Barrales C, Rodriguez-Tejero A, Martinez-Lopez A, Arias-Santiago S, Molina-Leyva A. Switching from Adalimumab Originator to Biosimilar: Clinical Experience in Patients with Hidradenitis Suppurativa. J Clin Med. 2022 Feb 15;11(4):1007. doi: 10.3390/jcm11041007. PMID: 35207280; PMCID: PMC8879480.

6- The following statement was added: Exclusion criteria were as follows: Age <18 years; inability or unwillingness to provide informed consent.

7- Figures modified

8- Thanks the reviewer for the point (8), probably the loss of efficacy by switching from originator to biosimilar could be explained by autoantibody production: In fact, it is known that neutralizing antibodies can be created during treatment with this class of biological drugs.23,24 Thus, antibodies could be potentiated in those patients who have undergone the switch, following antigenic stimulation caused by a similar but not homologous molecule.25

Reviewer 2 Report

Is this a sponsor-free study? If so, it must be emphasized. 

Each author must specify his/her conflict of interest.

Abstract. I strongly suggest to delete the last sentence ("These results seem to indicate that physicians should prescribe the originator...").

An important limitation is that no patient who began the therapy with the biosimilar switched to the originator.

Author Response

1- The following statement was added: This was a sponsor free retrospective multicenter study in which 14 Italian sites were involved. Patients were selected from March 2020 to June 2020.

2- Conflicts of Interest: The authors declare no conflicts of interest; this statement has been added to the manuscript.

3- Thanks to the reviewer for the suggestion, we deleted the sentence.

4- Thanks to the reviewer for the point, unfortunately we are aware of the limitation but italian legislation prevents us from switching easily from the biosimilar to the originator without clinical reason (loss of efficacy or adverse effects). So, in our clinical experience, the number of patients switched to the originator from the biosimilar wasn’t statistically significant.

Reviewer 3 Report

An interesting study.  The number of patients needs to increase. Was there a time gap between discontinuation of one drug and starting another? 

Author Response

We thank the reviewer for the suggestion, there wasn't any time gap between the discontinuation of one drug and the switch to another

Round 2

Reviewer 1 Report

The manuscript has improved and addressed the points raised during the first review. 

Reviewer 2 Report

I have no additional comments.